# The Reevaluation of Thrombin Time Using a Clot Waveform Analysis

**DOI:** 10.3390/jcm10214840

**Published:** 2021-10-21

**Authors:** Hideo Wada, Yuhuko Ichikawa, Minoru Ezaki, Takeshi Matsumoto, Yoshiki Yamashita, Katsuya Shiraki, Motomu Shimaoka, Hideto Shimpo

**Affiliations:** 1Department of General and Laboratory Medicine, Mie Prefectural General Medical Center, Yokkaichi 510-0885, Japan; katsuya-shiraki@mie-gmc.jp; 2Department of Central Laboratory, Mie Prefectural General Medical Center, Yokkaichi 510-0885, Japan; ichi911239@yahoo.co.jp (Y.I.); ajbyd06188@yahoo.co.jp (M.E.); 3Department of Transfusion Medicine and Cell Therapy, Mie University Hospital, Tsu 514-8507, Japan; matsutak@clin.medic.mie-u.ac.jp; 4Department of Hematology and Oncology, Mie University Graduate School of Medicine, Tsu 514-8507, Japan; yamayamafan4989@yahoo.co.jp; 5Department of Molecular Pathobiology and Cell Adhesion Biology, Mie University Graduate School of Medicine, Tsu 514-8507, Japan; motomushimaoka@gmail.com; 6Mie Prefectural General Medical Center, Yokkaichi 510-0885, Japan; hideto-shimpo@mie-gmc.jp

**Keywords:** CWA, thrombin time, platelet, thrombin burst

## Abstract

Object: Although thrombin burst has attracted attention as a physiological coagulation mechanism, clinical evidence from a routine assay for it is scarce. This mechanism was therefore evaluated by a clot waveform analysis (CWA) to assess the thrombin time (TT). Material and Methods: The TT with a low concentration of thrombin was evaluated using a CWA. We evaluated the CWA-TT of plasma deficient in various clotting factors, calibration plasma, platelet-poor plasma (PPP), and platelet-rich plasma (PRP) obtained from healthy volunteers, patients with thrombocytopenia, and patients with malignant disease. Results: Although the TT-CWA of calibration plasma was able to be evaluated with 0.01 IU/mL of thrombin, that of FVIII-deficient plasma could not be evaluated. The peak time of CWA-TT was significantly longer, and the peak height significantly lower, in various deficient plasma, especially in FVIII-deficient plasma compared to calibration plasma. The second peak of the first derivative (1st DP-2) was detected in PPP from healthy volunteers, and was shorter and higher in PRP than in PPP. The 1st DP-2 was not detected in PPP from patients with thrombocytopenia, and the 1st DP-2 in PRP was significantly lower in patients with thrombocytopenia and significantly higher in patients with malignant disease than in healthy volunteers. Conclusion: The CWA-TT became abnormal in plasma deficient in various clotting factors, and was significantly affected by platelets, suggesting that the CWA-TT may be a useful test for hemostatic abnormalities.

## 1. Introduction

It is well known that thrombin directly activates fibrinogen to generate fibrin formation [1]. Furthermore, thrombin activates many coagulation factors in the upper stream, such as clotting factor XI (FXI), FVIII, Factor X, and Factor V, resulting in thrombin generation to enhance coagulation reaction, a process known as thrombin burst [2,3]. The thrombin time (TT) is generally used to detect abnormalities of fibrinogen, such as dysfibrinogenemia [4,5], disseminated intravascular coagulation (DIC) [6], and liver dysfunction [7]. Therefore, the TT is used to measure fibrinogen concentrations [8]. In addition, it is also used to monitor anti-thrombin reagents [9]. Thrombin burst has generally been evaluated by thromboelastography (TEG) [10] and the thrombin generation test (TGT) [11,12]. 

The activated partial thromboplastin time (APTT) and prothrombin time (PT), which are inexpensive to conduct and allow for the easy performance of multiple assays, are established as routine assays for the coagulation system. However, where the APTT and PT reflect only a single dimension, such as the clotting time, the TEG and TGT are able to present results in two dimensions, including the time, width, height, or area. Unfortunately, these tests are expensive and time-consuming to perform compared to routine assays, at present. 

A clot waveform analysis (CWA)-APTT [13,14] and small-amount tissue factor (TF)-induced FIX activation (sTF/FIXa) assay [14,15] can show the peak time and peak height, allowing the evaluation of thrombin burst to be performed as easily as a routine assay. The CWA-APTT has been reported to detect very low levels of FVIII activity in patients with hemophilia A [16], and has proven useful for the differential diagnosis of hemophilia, acquired hemophilia A, lupus anticoagulant (LA), and DIC, as well as monitoring the results of anticoagulant therapy or bypass therapy in patients with FVIII inhibitors [17,18,19,20,21]. 

In the present study, a CWA using TT (CWA-TT) for physiological coagulation was used to investigate the mechanism underlying thrombin burst in calibration plasma and plasma deficient of various clotting factors. We also demonstrate the role of platelets in the coagulation system.

## 2. Materials and Methods

Platelet-rich plasma (PRP) and platelet-poor plasma (PPP) were collected from 12 patients with thrombocytopenia, 16 patients with malignant diseases, and 18 healthy volunteers (8 males and 12 females; 21 to 56 years old). The TT was measured using 0.5 IU thrombin (Thrombin 500 units, Mochida Pharmaceutical CO., LTD, Tokyo, Japan) with an ACL-TOP^®^ system (Instrumentation Laboratory, Bedford, MA, USA). Three types of curves are shown on this system monitor [19]. One shows the changes in the absorbance observed while measuring the TT, corresponding to the fibrin formation curve (FFC). The second is the first derivative peak of the absorbance (1st DP), corresponding to the coagulation velocity. The third is the second derivative peak of the absorbance (2nd DP), corresponding to the coagulation acceleration. FII-deficient plasma, FV-deficient plasma, FVII-deficient plasma, FVIII-deficient plasma, FIX-deficient plasma, FX-deficient plasma, FXI-deficient plasma, FXII-deficient plasma (Instrumentation Laboratory), and FXIII-deficient plasma (George King Bio-Medical Inc, Overland Park, KS, USA) were used as clotting factor-deficient plasma, and calibration plasma (Instrumentation Laboratory) was used as normal plasma. The fibrinogen concentrations in various deficient and calibration plasma were measured using a Thrombocheck Fib (L) (Sysmex, Kobe, Japan) and CS-5100 (Sysmex). 

PRP was prepared by centrifugation at 900 rpm for 15 min (platelet count, 40 × 10^10^/L in healthy volunteers), and PPP was prepared by centrifugation at 3000 rpm for 15 min (platelet count, <0.5 × 10^10^/L in healthy volunteers) [15].

### Statistical Analyses

The data are expressed as the median (25th to 75th percentile). Differences between PRP and PPP were examined for significance using Student’s *t*-test, and differences between independent groups were examined using the Mann–Whitney U-test. *p*-values of ≤0.05 were considered to indicate statistical significance. All statistical analyses were performed using the Stat flex software program (version 6. Artec Co Ltd., Osaka, Japan).

## 3. Results

The 2nd DP and 1st DP of calibration plasma using the CWA-TT with 0.01 IU/mL of thrombin were detected at 270 s, and the peak times of CWA-TT gradually shortened, whereas the peak heights of CWA-TT gradually increased, as the concentrations of thrombin increased (Figure 1). The 2nd DP and 1st DP of FVIII-deficient plasma using the CWA-TT with 0.01 to 0.1 IU/m of thrombin were not detected within 500 s, and the heights of the 2nd DP, 1st DP, and FFC of FVIII-deficient plasma using the CWA-TT with 0.5 to 1.0 IU/mL of thrombin were significantly lower than those of calibration plasma using the CWA-TT with the same concentration of thrombin. A total of 5 IU/mL of thrombin showed similar CWA-TT between calibration and FVIII-deficient plasma samples.

The heights of the 2nd DP, 1st DP, and FFC in FII-, FV-, FVII-, FVIII-, FIX-, FX-, FXI-, and FXII-deficient plasma using the CWA-TT with 0.5 IU thrombin were significantly lower than those in calibration plasma and FXIII-deficient plasma using the CWA-TT with 0.5 IU thrombin (Figure 2). In particular, the heights of the 2nd DP, 1st DP, and FFC in FVIII-deficient plasma were extremely low. The second peak of the 1st DP using the CWA-TT was observed only in calibration plasma, and FVII-, FXI-, FXII-, and FXIII-deficient plasma. 

The mean concentration of fibrinogen was 280 mg/dL in calibration plasma, 298 mg/dL in FII-, 313 mg/dL in FV-, 311 mg/dL in FVII-, 276 mg/dL in FVIII-, 310 mg/dL in FIX-, 295 mg/dL in FX-, 306 mg/dL in FXI-, and 334 mg/dL in FXII-deficient plasma.

The absorbances of FFC at 100 s in the mixing tests (*n* = 3) between calibration plasma and factor-deficient plasma showed dose dependence (Figure 3). The steep slope of the dose-dependent curve was inclined toward larger values in FVIII-deficient plasma, and toward smaller values in FXII-deficient plasma. The second peak of the 1st DP in normal plasma from healthy volunteers was significantly shorter and higher in PRP than in PPP (Figure 4). 

Regarding the analysis of the CWA-TT between PPP and PRP from healthy volunteers, the 2nd DPT and 1st DPT were significantly longer in PRP than in PPP, but the 1st DPT-2 and FFCT were significantly shorter in PRP than in PPP (Table 1). Although the 1st DPH-1 was significantly lower in PRP than in PPP, the 1st DPH-2 and FFCH were significantly higher in PRP than in PPP. The CWA-TT using PPP did not show a second peak of the 1st-derivative in patients with thrombocytopenia, although the CWA-TT using PRP did show a second peak of the 1st derivative. Using PPP for the CWA-TT, the 2nd DPT and 1st DPT-1 were significantly longer, and the 1st DPH-1 was significantly lower in patients with thrombocytopenia than in healthy volunteers. The 2nd DPH, 1st DPH-1, 1st DPH-2, and FFCH were significantly higher in patients with malignant disease than in healthy volunteers (Table 2). Using PRP for the CWA-TT, the 1st DPT-2 and FFCT were significantly longer in patients with thrombocytopenia than in healthy volunteers, and the FFCT was significantly shorter in patients with malignant disease than in healthy volunteers. The 1st DPH-2 was significantly lower in patients with thrombocytopenia than in healthy volunteers, and the 2nd DPH, 1st DPH-1, and 1st DPH-2 were significantly higher in patients with malignant diseases than in healthy volunteers (Table 3).

## 4. Discussion

The CWA-TT with FVIII-deficient plasma showed that a small amount of thrombin (≤0.1 IU/mL) failed to induce clot formation, suggesting that FVIII is required for physiological coagulation. As FVIII is reported to be markedly catalyzed and activated by thrombin [22], FVIII may play an important role in thrombin burst. Furthermore, the CWA-TT in plasma deficient of various clotting factors showed prolonged peak time, and decreased peak height. These findings suggest that various clotting factors are also required for the coagulation system involving thrombin burst induced by a small amount of thrombin [4,14]. However, CWA-TT reflects thrombin burst at thrombin concentrations ≤1.0 IU/mL, whereas at thrombin concentrations ≥5.0 IU/mL, CWA-TT strongly reflects the fibrinogen concentration [23]. In mixing texts to evaluate thrombin burst, a test with calibration plasma and FII-deficient plasma is useful as a control without thrombin burst, as FII-deficient plasma cannot cause a cycle of thrombin burst resulting in fibrin clot formation without thrombin burst. Therefore, a mixing test using CWA-TT proves that many coagulation factors, except for FXII, in which the millimeter absorbance was lower than that with FII-deficient plasma, may play an important role in thrombin burst.

In addition, the CWA-TT may be useful for evaluating the physiological and pathological coagulation induced by a small amount of thrombin. Physiological coagulation starts after small amounts of TF and FVIIa activate FIX, resulting in a small amount of thrombin. This thrombin activates not only fibrinogen, but also FV, FVIII, FIX, FX, and FXI, with the activation cycle from thrombin to FXI continuing for a short time thereafter [14]. The CWA-sTF/FIXa with a 2000-fold diluted PT reagent (recombinant TF) [24,25] was developed to evaluate physiological coagulation, and was shown to be capable of measuring the FVIII concentration. As a cross-mixing test of the CWA-TT between calibration plasma and FVIII-deficient plasma showed a good dose-response curve, the CWA-TT may be able to measure the FVIII concentration. 

The physiological coagulation system includes enhancement of clotting activation by phospholipids of platelets. However, most APTT reagents have some contact activation substance, and cannot demonstrate physiological coagulation [15]. Therefore, an sTF/FIXa assay [15,24] uses PRP as a physiological phospholipid instead of commercial APTT reagents. The CWA-TT also showed that the peak height, especially the 1st DPH-2, was higher in PRP than in PPP. The 1st DPH-2 was absent in PPP, and low in PRP. Activated FVII has been reported to generate thrombin in hemophilia via both platelet-dependent and platelet-independent mechanisms [26]. These findings therefore suggest that the thrombin burst mechanism may depend, at least partially, on platelets. However, the peak heights of the CWA-TT were extremely high, suggesting that patients with malignant diseases are in a hypercoagulable state. Further studies of the CWA-TT will be required to investigate its utility for the differential diagnosis of thrombocytopenia and hypercoagulability.

## 5. Conclusions

The peak time and height of CWA-TT became abnormal in plasma deficient of various clotting factors, and the CWA-TT was markedly affected by platelet counts, suggesting that the CWA-TT may be useful in testing for hemostatic abnormalities, such as thrombocytopenia or hypercoagulability. 

## Figures and Tables

**Figure 1 jcm-10-04840-f001:**
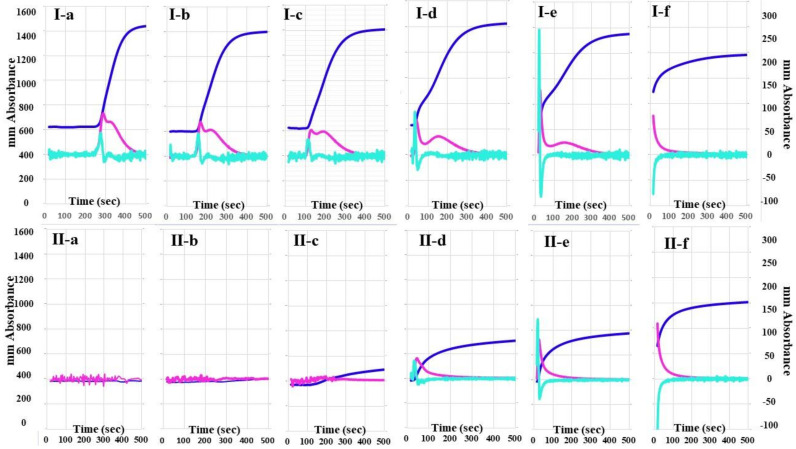
A clot waveform analysis for thrombin time. (**I**) Calibration plasma; (**II**) FVIII-deficient plasma; (**a**) thrombin 0.01 IU/mL; (**b**) thrombin 0.05 IU/mL; (**c**) thrombin 0.1 IU/mL; (**d**) thrombin 0.5 IU/mL; (**e**) thrombin 1.0 IU/mL; (**f**) thrombin 5.0 IU/mL. Navy line, fibrin formation curve; red line, 1st derivative curve (velocity); light blue, 2nd derivative curve (acceleration).

**Figure 2 jcm-10-04840-f002:**
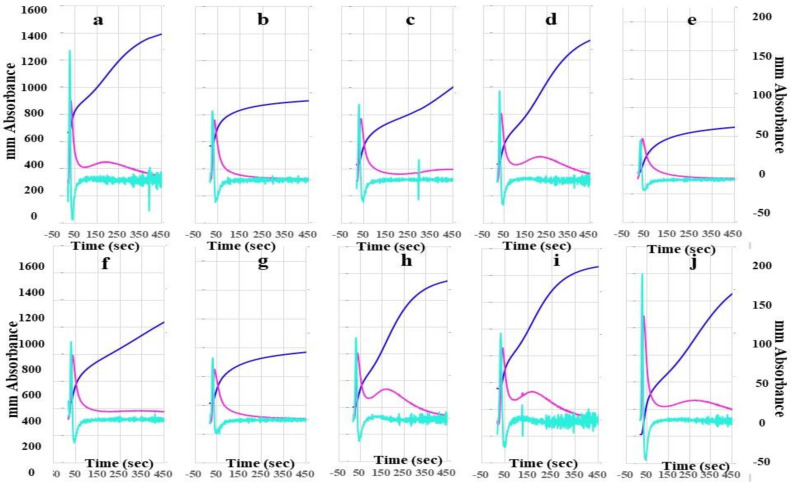
A clot waveform analysis for thrombin time. (**a**) Calibration plasma; (**b**) FII-deficient plasma; (**c**) FV-deficient plasma; (**d**) FVII-deficient plasma; (**e**) FVIII-deficient plasma; (**f**) FIX-deficient plasma; (**g**) FX-deficient plasma; (**h**) FXI-deficient plasma; (**i**) FXII-deficient plasma; (**j**) FXIII-deficient plasma; thrombin 0.5 IU/mL. Navy line, fibrin formation curve; red line, 1st derivative curve (velocity); light blue, 2nd derivative curve (acceleration).

**Figure 3 jcm-10-04840-f003:**
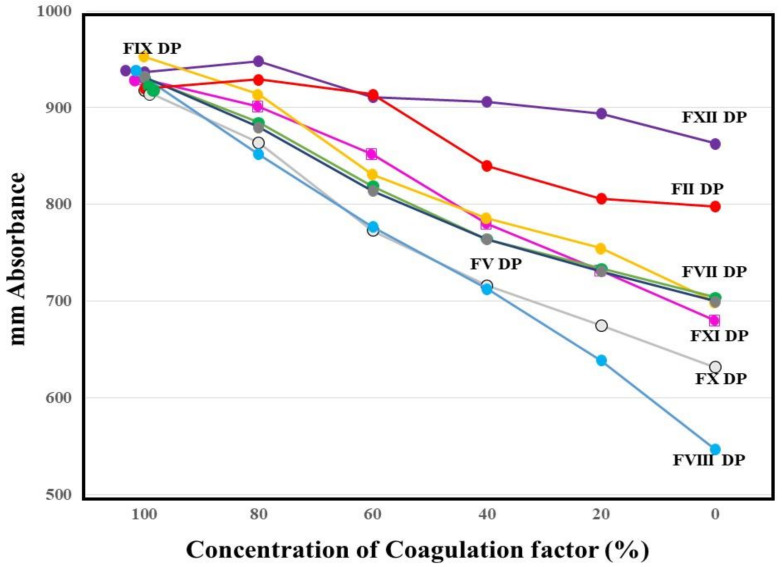
Mixing test between calibration plasma and plasma deficient of each factor using a clot waveform analysis for thrombin time (0.5 IU/mL of thrombin). The height of fibrin formation at 100 s was plotted. The mean values of three assays are shown. The standard deviation in each assay was less than 45 mm absorbance. DP, deficient plasma.

**Figure 4 jcm-10-04840-f004:**
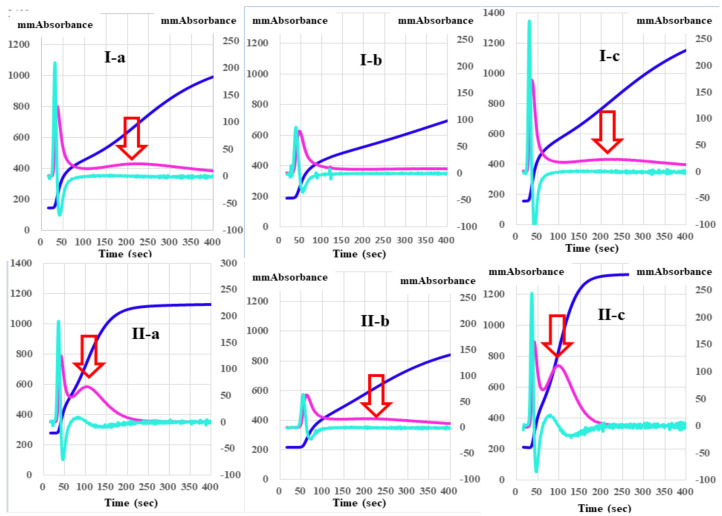
A clot waveform analysis of thrombin time (0.5 IU/mL) in platelet-poor plasma (**I**) and platelet-rich plasma (**II**) from healthy volunteers (**a**), patients with thrombocytopenia (**b**), and patients with malignant disease (**c**). Navy line, fibrin formation curve; red line, 1st derivative curve (velocity); light blue, 2nd derivative curve (acceleration); red arrow shows 1st DPT-2.

**Table 1 jcm-10-04840-t001:** Parameters of CWA-TT in PPP and PRP from healthy volunteers.

	2nd DPT (seconds)	2nd DPH (mm ABS)	1st DPT-1 (seconds)	1st DPH-1 (mm ABS)	1st DPT-2 (seconds)	1st DPH-2 (mm ABS)	FFCT (seconds)	FFCH (mm ABS)
PPP (*n* = 18)	30.1 *** (28.9–30.9)	164 (137–219)	36.3 ** (35.0–37.5)	113 * (96.5–132)	234 *** (220–278)	18.1 *** (15.4–22.9)	194 *** (185–204)	615 ** (560–664)
PRP (*n* = 18)	37.4 *** (33.1–43.3)	124 (110–180)	45.4 ** (39.2–50.2)	101 * (82.5–120)	134 *** (112–137)	45.3 *** (40.0–62.1)	127 *** (115–139)	730 ** (700–843)

Data are shown as the median (25–75 percentile). ***, *p* < 0.001, **, *p* < 0.01; *, *p* < 0.05 on comparing PPP and PRP. PPP, platelet-poor plasma; PRP (*n* = 18), platelet-rich plasma (*n* = 18); 2nd DPT, second derivative peak time; 2nd DPH, second derivative peak height; 1st DPT, first derivative peak time; 1st DPH, first derivative peak height, FFCT, fibrin formation curve time; FFCH, fibrin formation curve height; ABS, absorbance.

**Table 2 jcm-10-04840-t002:** Parameters of CWA-TT in PRP from healthy volunteers, patients with solid cancer, and patients with malignancy.

	2nd DPT (seconds)	2nd DPH (mm ABS)	1st DPT-1 (seconds)	1st DPH-1 (mm ABS)	1st DPT-2 (seconds)	1st DPH-2 (mm ABS)	FFCT (seconds)	FFCH (mm ABS)
Healthy volunteers (*n* = 18)	37.4 (33.1–43.3)	124 (110–180)	45.4 (39.2–50.2)	101 (82.5–120)	134 (112–137)	45.3 (40.0–62.1)	127 (115–139)	730 (700–842)
Thrombocytopenia (*n* = 12)	35.6 (32.3–38.7)	174 (98.6–219)	43.9 (39.9–44.7)	115 (73.6–140)	188 *** (146–216)	29.2 *** (26.6–34.4)	153 * (131–198)	763 (682–865)
Malignant diseases (*n* = 16)	33.7 (30.1–38.5)	268 ** (185–339)	39.3 (34.7–43.0)	154 *** (126–176)	111 (104–129)	82.2 *** (76.4–108)	102 * (98.2–118)	953 (813–1067)

Data are shown as the median (25–75 percentile). ***, *p* < 0.001, **, *p* < 0.01; *, *p* < 0.05 compared with healthy volunteers. PRP, platelet-rich plasma; 2nd DPT, second derivative peak time; 2nd DPH, second derivative peak height; 1st DPT, first derivative peak time; 1st DPH, first derivative peak height, FFCT, fibrin formation curve time; FFCH, fibrin formation curve height; ABS, absorbance.

**Table 3 jcm-10-04840-t003:** Parameters of CWA-TT in PPP from healthy volunteers, patients with solid cancer, and patients with malignancy.

	2nd DPT (seconds)	2nd DPH (mm ABS)	1st DPT-1 (seconds)	1st DPH-1 (mm ABS)	1st DPT-2 (seconds)	1st DPH-2 (mm ABS)	FFCT (seconds)	FFCH (mm ABS)
Healthy volunteers (*n* = 18)	30.5 (28.9–30.9)	164 (137–219)	36.3 (35.0–37.5)	113 (96.5–132)	234 (220–278)	18.1 (15.4–22.9)	194 (185–204)	615 (560–664)
Thrombocytopenia (*n* = 12)	34.9 * (31.0–39.2)	143 (78.5–255)	43.2 * (37.6–47.5)	106 * (70.2–157)	Not detectable	Not detectable	202 (152–209)	550 (492–637)
Malignant diseases (*n* = 16)	30.3 (28.6–46.1)	321 * (137–420)	35.0 (33.7–52.7)	166 *** (103–207)	228 (173–273)	28.5 ** (21.6–43.3)	187 (157–236)	738 ** (656–874)

Data are shown as the median (25–75 percentile). ***, *p* < 0.001, **, *p* < 0.01; *, *p* < 0.05 compared with healthy volunteers. PPP, platelet-poor plasma; 2nd DPT, second derivative peak time; 2nd DPH, second derivative peak height; 1st DPT, first derivative peak time; 1st DPH, first derivative peak height, FFCT, fibrin formation curve time; FFCH, fibrin formation curve height; ABS, absorbance.

## Data Availability

The data presented in this study are available on request to the corresponding author. The data are not publicly available due to privacy restrictions.

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
