# Peer review of "The Reevaluation of Thrombin Time Using a Clot Waveform Analysis"

_jcm, 2021, doi:10.3390/jcm10214840_

Round 1

Reviewer 1 Report

Please find below some comments and suggestions on your manuscript entitled “The reevaluation of thrombin time using a clot waveform analysis”.

First of all I have to say that clot waveform analysis (CWA) is still an emerging tool in clinical haemostasis diagnostics. By now it is not as widespread as thromboelastography and thrombin generation test. That is why accumulation of positive data on the use of CWA and its different modifications is a very essential task. In this respect I found your manuscript a valid contribution to further development of our knowledge of CWA capabilities.

On the other hand knowing that the field of CWA is relatively young and yet poorly standardized it is essential to make sure that all the experimental conditions are indicated detailed enough and all the date is represented in a most comprehensive manner.

  1. Thrombin concentration

Most of the previous studies on CWA were utilizing APTT test. In contrast your work focuses on thrombin time (TT) modification of CWA with clotting initiation by low thrombin concentrations. Up to my knowledge previously CWA for TT test was used only once in [Suzuki et al. Clot waveform analysis in Clauss fibrinogen assay contributes to classification of fibrinogen disorders //Thrombosis Research 174 (2019) 98–103], though using high thrombin concentrations and diluted plasma according to Clauss fibrinogen assay procedure.

As far as I’m concerned, a regular TT is almost insensitive to different clotting factors deficiencies (for example it is perfectly normal in hemophilia A and B), reflecting mostly the final stage of coagulation, i.e. conversion of fibrinogen to fibrin by thrombin. In this respect it seems that the results represented in your paper on different factor deficiencies could be obtained only if concentration of thrombin is way below the levels used in regular TT assay. I suggest adding some sort of the comparison between thrombin concentrations used in you study and in routine clinical TT assays.

  1. Potential influence of fibrin content on the results of mixing test

It is well known that absorbance observed in coagulation tests is very sensitive to fibrin content of plasma sample. It seems important to distinguish impact of fibrin content differences on the results of a mixing test from the impact of deficient coagulation factor under consideration. 

In other words if one will take two blood samples with full coagulation cascade (no coagulation deficiency) but with different fibrin concentrations and will perform a mixing test described in your paper he might obtain a clear linear dependency like those presented in Figure 3 simply due to fibrin concentration differences.

That is why it seems necessary to add some sort of control experiment confirming that there is no influence of fibrin differences on the dependencies observed in Figure 3.

  1. Mixing test results representation (Figure 3.)

Also in Figure 3 any sort of error bars are missing. Due to the selected form of data representation it is even hard to say how many experimental points (i.e. different concentrations) are presented for each dependency. All together it makes almost impossible to interpret these plots.

  1. Lastly in the very first sentence of the abstract you say: “Although thrombin burst has attracted attention as a physiological coagulation mechanism, clinical evidence for it is scarce”. Frankly speaking this statement is a confusing one. Since the introduction of thrombin generation test by Prof. Hemker and his colleagues there is vast evidence that thrombin burst is playing a pivotal role in coagulation in vivo. There are hundreds of papers (including clinical ones) studying this subject. So I’m not sure if for today it is correct to consider it as “scarce evidence”.

In summary I see the paper as a valid piece of research.

Author Response

First of all I have to say that clot waveform analysis (CWA) is still an emerging tool in clinical haemostasis diagnostics. By now it is not as widespread as thromboelastography and thrombin generation test. That is why accumulation of positive data on the use of CWA and its different modifications is a very essential task. In this respect I found your manuscript a valid contribution to further development of our knowledge of CWA capabilities.

On the other hand knowing that the field of CWA is relatively young and yet poorly standardized it is essential to make sure that all the experimental conditions are indicated detailed enough and all the date is represented in a most comprehensive manner.

  1. Thrombin concentration

Comments 1: Most of the previous studies on CWA were utilizing APTT test. In contrast your work focuses on thrombin time (TT) modification of CWA with clotting initiation by low thrombin concentrations. Up to my knowledge previously CWA for TT test was used only once in [Suzuki et al. Clot waveform analysis in Clauss fibrinogen assay contributes to classification of fibrinogen disorders //Thrombosis Research 174 (2019) 98–103], though using high thrombin concentrations and diluted plasma according to Clauss fibrinogen assay procedure.

As far as I’m concerned, a regular TT is almost insensitive to different clotting factors deficiencies (for example it is perfectly normal in hemophilia A and B), reflecting mostly the final stage of coagulation, i.e. conversion of fibrinogen to fibrin by thrombin. In this respect it seems that the results represented in your paper on different factor deficiencies could be obtained only if concentration of thrombin is way below the levels used in regular TT assay. I suggest adding some sort of the comparison between thrombin concentrations used in you study and in routine clinical TT assays.

Response 1.: Figure 1 shows the relationship between the thrombin concentration and CWA-TT. The following text has now been added.

“A total of 5 IU/ml of thrombin showed similar CWA-TT between calibration and FVIII deficient plasma samples.”

“However, CWA-TT reflects thrombin burst at thrombin concentrations ≤1.0 IU/ml, while at thrombin concentrations ≥5.0 IU/ml, CWA-TT strongly reflect the fibrinogen concentration [23].”

 As the above report is considered important, it was cited.

  1. Potential influence of fibrin content on the results of mixing test

Comments 2: It is well known that absorbance observed in coagulation tests is very sensitive to fibrin content of plasma sample. It seems important to distinguish impact of fibrin content differences on the results of a mixing test from the impact of deficient coagulation factor under consideration. 

In other words if one will take two blood samples with full coagulation cascade (no coagulation deficiency) but with different fibrin concentrations and will perform a mixing test described in your paper he might obtain a clear linear dependency like those presented in Figure 3 simply due to fibrin concentration differences.

That is why it seems necessary to add some sort of control experiment confirming that there is no influence of fibrin differences on the dependencies observed in Figure 3.

Response 2.: The fibrinogen concentration in each type of deficient plasm has now been mentioned in the Results. In mixing text to evaluate thrombin burst, a test with calibration plasma and FII-deficient plasma is useful as a control without thrombin burst, as FII-deficient plasma cannot cause a cycle of thrombin burst resulting in fibrin clot without thrombin burst. Therefore, a mixing test using CWA-TT proves that many clotting factors, except with FXII, may play an important role in thrombin burst.

  1. Mixing test results representation (Figure 3.)

Comments 3: Also in Figure 3 any sort of error bars are missing. Due to the selected form of data representation it is even hard to say how many experimental points (i.e. different concentrations) are presented for each dependency. All together it makes almost impossible to interpret these plots.

Response 3: The figure has now been revised for clarity. In addition, the following text was added instead of error bars. “The mean values of three assays were shown. The standard deviation in each assay was less than 45 mm absorbance.”

Comments 4: Lastly in the very first sentence of the abstract you say: “Although thrombin burst has attracted attention as a physiological coagulation mechanism, clinical evidence for it is scarce”. Frankly speaking this statement is a confusing one. Since the introduction of thrombin generation test by Prof. Hemker and his colleagues there is vast evidence that thrombin burst is playing a pivotal role in coagulation in vivo. There are hundreds of papers (including clinical ones) studying this subject. So I’m not sure if for today it is correct to consider it as “scarce evidence”.

Response 4.: The sentenceclinical evidence for it is scarce.” was changed to “clinical evidence from a routine assay for it is scarce.” The report by Hemker et al was cited.

In summary I see the paper as a valid piece of research.

Reviewer 2 Report

Major revision

I am very interested in the manuscript describe by a clot waveform analysis (CWA) with small amounts of thrombin.

I was especially excited about the high sensitivity to FVIII. In addition, it can contribute to the interpretation of platelet-related diseases.

I thought that we could observe the evaluation of coagulation function, which is different from the conventional concept of separating the extrinsic and intrinsic pathways of a prothrombin time (PT) and an activated partial thromboplastin time (APTT).

I have a few more questions and would be very grateful if they could be resolved.

Minor revision

・p2.L80 A platelet count of 40 x 1010/L in PRP would predict a strong turbidity of the plasma. What is the reason behind this platelet count? Also, have you been able to keep this platelet count in patients with thrombocytopenia?

・p2.L93 The clot waveform analysis is dependent on fibrinogen values. Are there any significant differences in the value of fibrinogen in each factor deficient plasma and healthy volunteers? We are especially interested in the fibrinogen value of the factor VIII deficient plasma used for the measurements that showed the lower levels.

・p.3L106 I am not certain about the number of plots for the mixing test in Figure 3. I would like you to add the number of points. Also, in the mixing test, did the absorbance increase from the point of low normal plasma mixed volume?

・p.5 I don't find any explanation for (c) in Figure 4, I-c, II-c.

・p.5 How many measurements have you done with PRP with Table1 and Table2?

I thought there was not enough description of reproducibility not only for this assay but also for other methods.

・p.6 L162 What is the reason why only FVIII was decreased in CWA-TT and other coagulation factors were not affected significantly? Do you consider this to be caused by the concentration of thrombin?

Author Response

Comments and Suggestions for Authors

Major revision

Comments 1: I am very interested in the manuscript describe by a clot waveform analysis (CWA) with small amounts of thrombin.

Response 1: Thank you very much. We intend to further our studies in the next step.

Comments 2: I was especially excited about the high sensitivity to FVIII. In addition, it can contribute to the interpretation of platelet-related diseases.

Response 2: Thank you very much. We will proceed with further work in the fields of hemophilia and ITP

Comments 3: I thought that we could observe the evaluation of coagulation function, which is different from the conventional concept of separating the extrinsic and intrinsic pathways of a prothrombin time (PT) and an activated partial thromboplastin time (APTT).

Response 3: Thank you very much. We will conduct further studies under various conditions.

I have a few more questions and would be very grateful if they could be resolved.

Minor revision

Comments 4: ・p2.L80 A platelet count of 40 x 1010/L in PRP would predict a strong turbidity of the plasma. What is the reason behind this platelet count? Also, have you been able to keep this platelet count in patients with thrombocytopenia?

Response 4: The relationship between the platelet count and a CWA-has been reported in Reference 14. Therefore, “in healthy volunteers” and [14] were added to this sentence.

Comments 5: p2.L93 The clot waveform analysis is dependent on fibrinogen values. Are there any significant differences in the value of fibrinogen in each factor deficient plasma and healthy volunteers? We are especially interested in the fibrinogen value of the factor VIII deficient plasma used for the measurements that showed the lower levels.

Response 5: The fibrinogen concentration in each type of deficient plasma has anow been mentioned in the Results.

Comments 6:・p.3L106 I am not certain about the number of plots for the mixing test in Figure 3. I would like you to add the number of points. Also, in the mixing test, did the absorbance increase from the point of low normal plasma mixed volume?

Response 6: “(n=3)” was added. “The mean values of three assays were shown,” was added to the legend of Figure 3. Unfortunately, the meaning of your second question is not clear to me. The absorbance increased as the ratio of normal plasma to deficient plasma increased.   

Comments 7:・p.5 I don't find any explanation for (c) in Figure 4, I-c, II-c.

Response 7: “(c)” was added to the legend of Figure 4.

Comments 8:・p.5 How many measurements have you done with PRP with Table1 and Table2? I thought there was not enough description of reproducibility not only for this assay but also for other methods.

Response 8: Healthy volunteer and patient numbers have now been described in the Materials and Methods. In addition, “n=18” was added to the legend of Table 1. “n” was added to Table 2 and 3. As our experiments were performed three times and patient assays were performed twice, the reproducibility was confirmed.

Comments 9:・・p.6 L162 What is the reason why only FVIII was decreased in CWA-TT and other coagulation factors were not affected significantly? Do you consider this to be caused by the concentration of thrombin?

Response 9: “As FVIII is reported to be markedly catalyzed and activated by thrombin [22], FVIII may play an important role in thrombin burst.” was added to the Discussion. The thrombin concentration was stated on page 6, lines 174-175.  
